# CYCLE MONOTONICITY OF ADVERSARIAL ATTACKS FOR OPTIMAL DOMAIN ADAPTATION

## ABSTRACT

We reveal an intriguing connection between adversarial attacks and cycle monotone maps, also known as optimal transport maps. Based on this finding, we developed a novel method named *source fiction* for semi-supervised optimal transport-based domain adaptation. In our algorithm, instead of mapping from target to the source domain, optimal transport maps target samples to the set of adversarial examples. The trick is that these adversarial examples are labeled target samples perturbed to look like source samples for the source domain classifier. Due to the cycle monotonicity of adversarial attacks, optimal transport can naturally approximate this transformation. We conduct experiments on various datasets and show that our method can notably improve the performance of optimal transport methods in semi-supervised domain adaptation.

## 1 INTRODUCTION

Optimal Transport (OT) is a powerful framework to solve mass moving problems for probability distributions. Over the past decades, it has been successfully applied in mathematics (Ferradans et al., 2014), economics (Reich, 2013) and machine learning (Arjovsky et al., 2017; Mroueh, 2019; Solomon et al., 2015; Colombo et al., 2021). The key property of OT maps is a cycle monotonicity (McCann, 1995), such maps cannot be improved; it is impossible to perturb it and get something more economical (Villani, 2008, §5).

In machine learning, OT finds a rich application in domain adaptation (DA) problem (Courty et al., 2015; Perrot et al., 2016; Rakotomamonjy et al., 2020). In DA, the task is to learn a model $f_\theta$, e.g., a classifier, from a source domain $\Omega_s$ to perform well on a different (related) target domain $\Omega_t$ (Ben-David et al., 2010a). The conventional OT approach for domain adaptation is to use OT map $\Omega_t \to \Omega_s$ to transform target domain $\Omega_t$ to source $\Omega_s$ and then apply the model $f_\theta$ to mapped samples.

With few labeled samples in target domain $\Omega_{semi}$ we can apply semi-supervised domain adaptation (Wang & Deng, 2018). For semi-supervised OT, labels are used to prevent the transportation of the target sample into the source sample from another class (Courty et al., 2016; Yan et al., 2018). But labels do not provide supervision for the map between two domains because supervision involves input-output pairs, and labels do not give such pairwise matching between source and target samples. Moreover, supervision for OT has to involve a cycle monotone matching between source and target samples and save the class-wise structure.

In our paper, we propose an algorithm we call *source fiction* that provides cycle monotone supervision to OT and improves the performance of OT in semi-supervised DA. The core of this algorithm is our finding that adversarial attacks (Szegedy et al., 2014), more precisely Iterative Fast Sign Gradient Descent (Goodfellow et al., 2014, FSGD) are cycle monotone transformations over the dataset with quadratic cost.

We propose to replace source domain $\Omega_s$ with domain $\Omega_f$ that is constructed by the adversarial attack on labeled samples in target domain $\Omega_{semi}$ to be accurately classified by source domain classifier $f_\theta$. Since domain $\Omega_f$ is a cycle monotone transformation of the target domain $\Omega_{semi}$, it becomes natural to use $\Omega_{semi} \to \Omega_f$ as supervision for OT mapping and then use the resulted map to transport from target domain without labels to $\Omega_f$.

**The main contributions** of this paper are: (i) We prove that an adversarial attack produces cycle monotone transformation of the dataset (Section 3); (ii) We propose a novel *source fiction* algorithm for OT-based semi-supervised domain adaptation. The algorithm incorporates the adversarial attack to the conventional pipeline of OT for DA (Section 4).

## 2 BACKGROUND AND RELATED WORK

### 2.1 ADVERSARIAL ATTACKS

Adversarial examples are samples that are similar to the true samples $D(x, x') \leq \epsilon$, but "fool" a selected classifier and tend it to make incorrect predictions $\text{argmax} \, p(y \mid x') \neq y_{\text{true}}$ (Szegedy et al., 2014). The phenomenon of the vulnerability of machine learning models to adversarial examples breeds a great deal of concern in many learning scenarios (Papernot et al., 2017; Yuan et al., 2019; Schott et al., 2019; Xie et al., 2017).

A large body of work on adversarial attacks exists; in our paper, we consider Fast Sign Gradient Descent (FSGD) by Goodfellow et al. (2014); iterative version of FSGD can be presented as

$$x'_0 = x, \quad x'_{i+1} = \text{clip}_{x,\epsilon} \left\{ x'_i - \alpha \, \text{sign} \left( \nabla_x L \left( \theta, x'_i, y \right) \right) \right\} \tag{1}$$

With sample $x$, the target label $y$, and classifier $f$ with parameters $\theta$, we can obtain adversarial examples using gradient descent, by solving the min-max problem, maximizing the loss $L$ and minimizing perturbation for a sample $x$ in some class $y$ with respect to some perturbation size $\epsilon$.

Newly, various properties of adversarial examples were studied (Petrov & Hospedales, 2019; Papernot et al., 2016; Ilyas et al., 2019), and applications of adversarial examples for models accuracy improvements were proposed (Xie et al., 2019; Yang et al., 2020).

Previously connection of OT and adversarial examples were studied in the context of robustness problems (Pydi & Jog, 2020; Bouniot et al., 2021). Wong et al. (2019) proposed the idea of Wasserstein adversarial attack with Sinkhorn iterations; this algorithm allows to find adversarial perturbations with respect to Wasserstein ball.

### 2.2 OPTIMAL TRANSPORT

OT aims at finding a solution to transfer mass from one distribution to another with the least effort. Monge's problem was the first example of the OT problem and can be formally expressed as follows:

$$\inf_{T\#\mu=\nu} \int_{\Omega_\mu} c(\mathbf{x}, T(\mathbf{x}))\mu(\mathbf{x})d\mathbf{x} \tag{2}$$

The Monge's formulation of OT aims at finding a mapping $T : \Omega_\mu \to \Omega_\nu$ of the two probability measures $\mu$ and $\nu$ and a cost function $c : \Omega_\mu \times \Omega_\nu \to \mathbb{R}_+$, where $T\#\mu_s = \nu_t$ represents the mass preserving push forward operator. In Monge's formulation, $T$ cannot split the mass from a single point. The problem is that with such constraints, the mapping $T$ may not even exist.

To avoid this problem, Kantorovitch proposed a relaxation (Villani, 2008). Instead of obtaining a mapping, the goal is to seek a joint distribution over the source and the target that determines how the mass is allocated. For a given cost function $c : \Omega_\mu \times \Omega_\nu \to \mathbb{R}_+$, the primal Kantorovitch formulation can be expressed as the following problem:

$$\min_{\gamma \in \Pi(\mu,\nu)} \left\{ \int_{\Omega_\mu \times \Omega_\nu} c(\mathbf{x}, \mathbf{y})d\gamma(\mathbf{x}, \mathbf{y}) = \mathbb{E}_{(\mathbf{x},\mathbf{y})\sim\gamma}[c(\mathbf{x}, \mathbf{y})] \right\} \tag{3}$$

In primal Kantorovitch formulation, we look for a joint distribution $\gamma$ with $\mu$ and $\mu$ as marginals that minimize the expected transportation cost. If the independent distribution $\gamma(\mathbf{x}, \mathbf{y}) = \mu(\mathbf{x})\nu(\mathbf{y})$ respects the constraints, linear program is convex and always has a solution for a semi-continuous $c$:

$$\Pi(\mu, \nu) = \left\{ \gamma \in P(\Omega_\mu, \Omega_\nu) : \int \gamma(\mathbf{x}, \mathbf{y})d\mathbf{y} = \mu(\mathbf{x}), \int \gamma(\mathbf{x}, \mathbf{y})d\mathbf{x} = \nu(\mathbf{y}) \right\} \tag{4}$$

The primal Kantorovitch formulation can be also presented in dual form as stated by the Rockafellar-Fenchel theorem (Villani, 2008):

$$\max_{\phi \in \mathcal{C}(\Omega_\mu), \psi \in \mathcal{C}(\Omega_\nu)} \left\{ \int \phi d\mu + \int \psi d\nu \mid \phi(\mathbf{x}) + \psi(\mathbf{y}) \leq c(\mathbf{x}, \mathbf{y}) \right\} \quad (5)$$

After finding a solution to the transport problem, OT provides a measure of dissimilarity between the two distributions. This similarity is also called the Wasserstein distance (Villani, 2008):

$$W_p(\mu_s, \nu_t) = \min_{\gamma \in \Pi(\mu, \nu)} \left\{ \int_{\Omega_\mu \times \Omega_\nu} c(\mathbf{x}, \mathbf{y}) d\gamma(\mathbf{x}, \mathbf{y}) \right\}^{\frac{1}{p}} \quad (6)$$

where $c(\mathbf{x}, \mathbf{y}) = \|\mathbf{x} - \mathbf{y}\|^p$ and $p > 1$. The Wasserstein distance encodes the geometry of the space through the optimization problem and can be used on any distribution of mass.

The main geometric property of the OT is cycle monotonicity. Informally, a c-cycle monotone plan is a plan that cannot be improved, for all points $x_0 \ldots x_i, y_0 \ldots y_i$ holds:

$$\sum_{i=1}^N c(\mathbf{x}_i, \mathbf{y}_i) \leq \sum_{i=1}^N c(\mathbf{x}_i, \mathbf{y}_{i+1}) \quad (7)$$

It is impossible to perturb it and get something more economical (Villani, 2008, §5).

## 2.3 OPTIMAL TRANSPORT FOR DOMAIN ADAPTATION

**Domain Adaptation:** DA techniques try to make the target domain $\Omega_t$ samples $X_t = (x_i^t)_{i=1}^{N_t}$ be closer to the source domain $\Omega_s$ samples $X_s = (x_i^s)_{i=1}^{N_s}$ to make the source classifier $f_\theta$ perform accurately on the target domain (Ben-David et al., 2010a;b; Germain et al., 2013). In semi-supervised settings, target domain $\Omega_{semi}$ consists of unlabeled samples $X_t = (x_i^t)_{i=1}^{N_t}$ and small number of samples $X_l = (x_i^l)_{i=1}^{N_l}$ associated with labels $Y_l = (y_i^l)_{i=1}^{N_l}$. These labels can be used to train DA algorithm match target samples with source samples that have the same labels.

Different discrete OT algorithms can be applied for domain adaptation (Courty et al., 2015). For example, Earth Mover's Distance (EMD) (Courty et al., 2015; Flamary et al., 2021) solves the most typical primal Kantorovich OT problem (equation 4). This method seeks an optimal coupling $\gamma$, which minimizes the displacement cost between two domains with respect to some distance.

Sinkhorn algorithm (Cuturi, 2013) and its variations with a group lasso regularization (L1L2) and Laplacian regularization (L1LP) (Courty et al., 2015) are OT algorithms that apply lightspeed computation of OT in DA. Perrot et al. (2016) proposed Linear OT mapping estimators (MapT) that jointly learn the coupling $\gamma$ equation 4 and transport map $T$ linked to the original Monge problem equation 2.

In semi-supervised DA with OT, labels are used to penalty the transport $\Omega_{semi}(\gamma) = \langle \gamma, \mathbf{M} \rangle$ by a $n_s \times n_t$ cost matrix $M\rangle$ where $M(i, j) = 0$ when $y_i^s = y_j^t$ and $+\infty$ otherwise (Courty et al., 2016; Yan et al., 2018). In domain adaptation with label and target shift, OT was used to align probability distributions even for a few domains (Redko et al., 2019; Rakotomamonjy et al., 2020). Most of these algorithms are presented in POT library (Flamary et al., 2021), which provides state-of-the-art OT algorithms to solve the domain adaptation.

**Discrete OT:** Different discrete OT algorithms can be applied for domain adaptation (Courty et al., 2015). For example, Earth Mover's Distance (EMD) (Courty et al., 2015; Flamary et al., 2021) solves the most typical primal Kantorovich OT problem (equation 4). This method seeks an optimal coupling $\gamma$, which minimizes the displacement cost between two domains with respect to some distance.

**Neural OT:** The connection between OT and deep networks was proposed for unsupervised domain adaptation (Damodaran et al., 2018) and representation transfer between teacher and student in transfer learning (Li et al., 2020). It was also shown that mini-batch learning that is correctly applied in deep learning can be employed to OT for efficient measure between data distributions (Fatras et al., 2021).

Recently, there has been a solid push to incorporate Input Convex Neural Networks (ICNNs) (Amos et al., 2017) in OT problems. According to Rockafellar's Theorem (Rockafellar, 1966), every cyclical monotone mapping $g$ is contained in a sub-gradient of some convex function $f : \mathcal{X} \to \mathbb{R}$. Furthermore, according to Brenier's Theorem (Theorem 1.22 of Santambrogio (2015)), these gradients uniquely solve the Monge problem equation 2. Following these theorems, a range of approaches explored ICNNs as parameterized convex potentials in dual Kantorovich problem equation 5 (Taghvaei & Jalali, 2019; Makkuva et al., 2020).

Further development of this approach enabled the construction of the non-minimax Wasserstein-2 generative framework (Korotin et al., 2019) that can solve domain adaptation and Wasserstein-2 Barycenters estimation (Fan et al., 2020; Korotin et al., 2021b). Compared to discrete OT, neural methods provide generalized OT methods that can ensure out-of-sample estimates.

In our paper, in comparison to this method, we do not propose a new domain adaptation algorithm, new OT algorithm, or adversarial attack and defense based on OT. We presented a connection between OT and adversarial attacks and showed how this connection can be used for semi-supervised OT.

## 3    CYCLE MONOTONICITY OF ADVERSARIAL ATTACKS

This section demonstrates the cyclical monotonicity of adversarial attacks. We prove that with mild assumptions on the attack, it is cycle monotone w.r.t. the quadratic cost $c(x, y) = \frac{1}{2}\|x - y\|^2$.

**Lemma 1** (Cycle monotonicity of small perturbations of a dataset.)**.** *Let $x_1, \ldots, x_N \in \mathbb{R}^D$ be a dataset of $N$ distinct samples. Let $x'_1, \ldots, x'_N$ be its $\leq \epsilon$-perturbation, i.e. $\|x_n - x'_n\| \leq \epsilon$ for all $n = 1, 2, \ldots, N$. Assume that $\epsilon \leq \frac{1}{2} \min\limits_{n_1, n_2} \|x_{n_1} - x_{n_2}\|$, i.e. the perturbation does not exceed $\frac{1}{2}$ of the minimal pairwise distance between samples. Then for all $K$ and $\overline{1, N}$ it holds:*

$$\sum_{k=1}^{K} \frac{1}{2}\|x_{n_k} - x'_{n_k}\|^2 \leq \sum_{k=1}^{K} \frac{1}{2}\|x_{n_k} - x'_{n_{k+1}}\|^2 \tag{8}$$

*i.e. set $(x_1, x'_1), \ldots, (x_N, x'_N)$ or, equivalently, the map $x_k \mapsto x_{k'}$ is cycle monotone.*

*Proof.* Due to triangle inequality for $\| \cdot \|$, we have

$$\|x_{n_k} - x'_{n_{k+1}}\| \geq \underbrace{\|x_{n_k} - x_{n_{k+1}}\|}_{\geq 2\epsilon} - \underbrace{\|x_{n_{k+1}} - x'_{n_{k+1}}\|}_{\leq \epsilon} = \epsilon. \tag{9}$$

Taking the square of both sides and summing equation 9 for $k = 1, 2, \ldots, K$ yields

$$\sum_{k=1}^{K} \|x_{n_k} - x'_{n_{k+1}}\|^2 \geq \sum_{k=1}^{K} \epsilon^2 = K\epsilon^2. \tag{10}$$

Due to the assumptions of the lemma, the following inequality holds true:

$$\sum_{k=1}^{K} \|x_{n_k} - x'_{n_k}\|^2 \leq \sum_{k=1}^{K} \epsilon^2 \leq K\epsilon^2. \tag{11}$$

We combine equation 10 and equation 11 to obtain

$$\sum_{k=1}^{K} \|x_{n_k} - x'_{n_k}\|^2 \leq \sum_{k=1}^{K} \|x_{n_k} - x'_{n_{k+1}}\|^2,$$

which is equivalent to

$$\sum_{k=1}^{K} c(x_{n_k}, x'_{n_k}) \leq \sum_{k=1}^{K} c(x_{n_k}, x'_{n_{k+1}}), \tag{12}$$

and yield cycle monotonicity w.r.t. quadratic cost $c(x, y) = \frac{1}{2}\|x - y\|^2$.    □

Adversarial attacks are small perturbations of the dataset, therefore, we immediately obtain

**Corollary 1.** *Let $x_1, \ldots, x_N \in \mathbb{R}^D$ be a dataset of $N$ distinct samples. Then any adversarial attack $x_n \mapsto x'_n$ on the dataset with $\epsilon \leq \frac{1}{2} \min\limits_{n_1, n_2} \|x_{n_1} - x_{n_2}\|$ is cycle monotone.*

Corollary 1 suggests using the *optimal* map to transform target domain $\Omega_t$ to domain $\Omega_f$ formed by the cycle-monotone adversarial attack.

## 4    SOURCE FICTION ALGORITHM

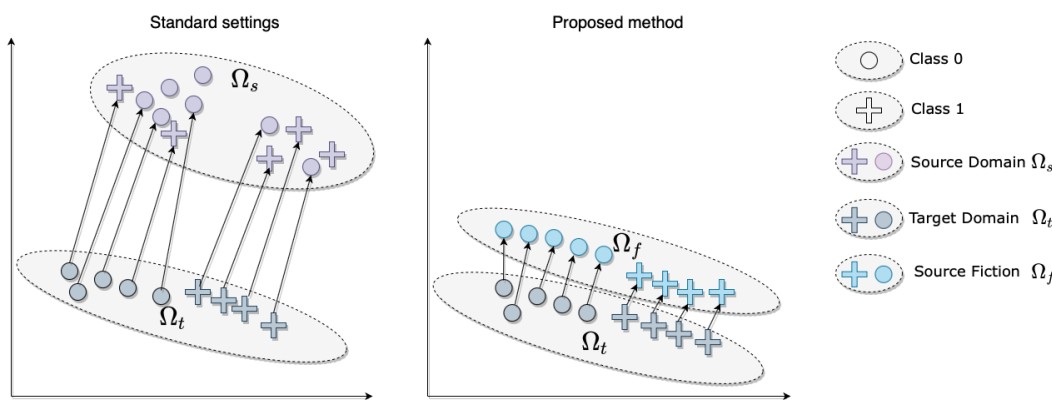

Figure 1: Visual example of adaptation with source fiction. Each object in target domain is closest to the corresponding object in source fiction domain in terms of some distance.

---

**Algorithm 1:** Source fiction for optimal domain adaptation

---

**Input:** Classifier $f_\theta$, optimal transport algorithm OT, source $\Omega_s$, labeled $X_l$ and unlabeled $X_t$ samples in target domain $\Omega_{semi}$, empty set $X_f$, perturbations size $\epsilon$

**Initialize:** $\epsilon \leq \frac{1}{2} \min\limits_{n_1, n_2} \|x_{n_1} - x_{n_2}\|$ for $x_n$ in $X_l$

Pretrain classifier $f_\theta$ on source $\Omega_s$ domain $X_s Y_s$ pairs.

**for**  x, y in $(X_l, Y_l)$ **do**

    **for** steps $n, \ldots, N$ **do**

        $x'_0 = x, \quad x'_{n+1} = \text{clip}_{x,\epsilon} \{x'_n - \alpha \, \text{sign} \left( \nabla_x L \left( \theta, x'_n, y \right) \right) \}$

    **end for**

    Append $x'$ examples to $X_f$

**end for**

**Return:** Domain $\Omega_f$

Find a map $\Omega_{semi} \to \Omega_f$ using OT.

Apply the classifier $f_\theta$ to samples $X_f$ mapped from $X_t$.

---

In the previous section, we considered that adversarial attack applies cycle-monotone transformation over the dataset. In this section, we propose an application of this property for DA. To manage this property, we propose to replace the source domain with the domain that is cycle monotone to the target domain, and at the same time, accurately classifying by source domain classifier. Due to the vulnerability of machine learning models to perturbation, we can turn any sample into accurately classified in practice, in the same way we create adversarial examples.

In our method, we propose to use a targeted FSGD adversarial attack (equation 1) to source domain classifier $f_\theta$, with target label $y$ equal to the true class of the given sample. Such attack adds to the image features of the class to which the image really belongs (Ilyas et al., 2019). Applying attack using labeled target samples $X_l = (x_i^l)_{i=1}^{N_l}$ with $y$ value in equation 1 equal to the true class, we obtain a new domain $\Omega_f$ with samples $X_f = (x_i^f)_{i=1}^{N_l}$ that are examples of cycle monotone transformation of the target domain samples, and at the same time accurately classifying by source

domain classifier $f_\theta$. Following corollary 1 to obtain monotonicity, the size of perturbation $\epsilon$ is set $\leq \frac{1}{2} \min_{n_1, n_2} \|x_{n_1} - x_{n_2}\|$ for all $x_n$ in $X_l$.

As a result, for each sample in the labeled target domain, we obtain a corresponding sample in $\Omega_f$. While $\Omega_f$ is cycle monotone to the target domain, we have a low quadratic cost between each target sample and its corresponding sample in $\Omega_f$ and a higher quadratic cost between this sample and all other samples in $\Omega_f$. To apply adaptation of classifier $f_\theta$ on the target domain samples without labels $\Omega_t$, we use OT to find a map between $\Omega_{semi}$ to $\Omega_f$, and then apply transport on samples without labels $\Omega_t$.

Figure 1 demonstrates the difference between standard DA by OT and DA using *source fiction*. The pipeline of learning OT for DA using *source fiction* $\Omega_f$ is presented in Algorithm 1.

## 5 EXPERIMENTS

In this section, we present details and the results of the experiments. The goal of our experiments is to demonstrate that our algorithm can improve the performance of fundamental discrete OT algorithms and novel neural networks based OT methods.

### 5.1 DATASETS

**Digits datasets:** We evaluated our method on Digits datasets MNIST (LeCun & Cortes, 2010), USPS (Hull, 1994), SVNH (Netzer et al., 2011), and MNIST-M (Ganin & Lempitsky, 2015). The experiments are conducted using two settings on each dataset: (1) optimizing OT using available labels to find a map between target and source domains and (2) optimizing OT to find a map between target and *source fiction* domains. Each dataset consists of 10 classes of digits images with different numbers of train and test samples.

**Modern Office-31 dataset:** Besides the Digits dataset that consists of only ten classes in each domain, we tested the performance of OT with the *source fiction* method on the Modern Office-31 dataset (Ringwald & Stiefelhagen, 2021). The Modern Office-31 dataset is one of the most extensive and diverse datasets for domain adaptation, with 31 classes in three domains: Amazon (A), Synthetic (S), and Webcam (W). Compared to the Digits and original Office-31 dataset (Saenko et al., 2010), this dataset includes synthetic to real transfer tasks, which is problematic. In our experiments, we used DLSR (D) domain from the original Office-31 to properly estimate the proposed algorithm.

### 5.2 SETTINGS

**Source classifier:** For the source domain classifier, we trained ResNet50 (He et al., 2016) to achieve $90+$ accuracy on the test set of each domain in Digits and Modern Office-31 dataset. The classifier was trained using Adam (Ruder, 2016) optimizer with 1e-3 learning rate, the size of latent space before the output layer was set equal to $2048$. After training, we applied domain adaptation by moving mass in the latent space of the source classifier.

**Discrete OT:** We tested several OT algorithms that apply semi-supervised adaptation: EMD, Sinkhorn, SinkhornL1L2, SinkhornLPL2, and MapT. For experiments, we used the POT library with quadratic cost distance in each method. Regularization size was equal to 4 for Sinkhorn, SinkhornL1L2, SinkhornLPL2, and MapT; all other hyperparameters were similar to the default, presented in POT. For *source fiction*, we used 50 steps FSGD with $\epsilon$ equal to 0.45, we find that such value allows to achieve strong perturbations and at the same time mostly satisfies $\epsilon \leq \frac{1}{2} \min_{n_1, n_2} \|x_{n_1} - x_{n_2}\|$ requirement for all domains. The results are presented in Table 1. All results are averaged over ten runs with different sets of labeled samples in the target domain. The top part of the table represents standard settings the bottom part presents results using *source fiction*. It is important to note that for mapping target to source, available target domain labels were used in OT solvers to apply regularization (Courty et al., 2015, §5).

**Neural OT:** Following recent benchmarking results of neural OT (Korotin et al., 2021a), we choose two methods to apply domain adaptation: W2GN (Korotin et al., 2019) and MM:R (Nhan Dam et al., 2019; Korotin et al., 2021a). As potentials $\phi$ and $\psi$ in neural OT methods, we used Dense

ICNN (Korotin et al., 2019) with three hidden layers [64, 64, 32] for W2GN and MM:R methods. Potentials were pertained to apply invariant transformation using Adam optimizer with $lr$ equal to 1e-4. For Wasserstein's objective, potentials were trained 300 epochs with Adam optimizer and $lr$ equal to 1e-3. The results are presented in Table 2, the top part of the table represents standard settings with 10 and 100 known labels per class, and the bottom part presents results using *source fiction*. In setting with the target to source mapping, labels were used to build class-wise pairs.

| METHOD | MNIST SVHN | SVHN MNIST | MNIST USPS | USPS MNIST | MNIST M-MNIST |
|---|---|---|---|---|---|
| SOURCE | 22.0 | 79.0 | 74.1 | 87.37 | 33.56 |
| EMD | 21.2 | 68.7 | 79.2 | 79.48 | 56.1 |
| SINKHORN | 21.8 | 68.8 | 82.1 | 81.2 | 55.7 |
| SINKHORNLPL1 | 21.8 | 68.8 | 84.8 | 82.73 | 55.7 |
| SINKHORNL1L2 | 21.8 | 68.8 | 84.8 | 82.73 | 55.7 |
| MAPOT | 21.8 | 69.9 | 84.1 | 84.1 | 62.3 |
| EMD | **23.0** | **86.3** | **83.1** | **92.8** | **62.7** |
| SINKHORN | **25.5** | **86.2** | **83.8** | **92.8** | **62.9** |
| SINKHORNLPL1 | **25.5** | **86.3** | **88.3** | **93.0** | **63.0** |
| SINKHORNL1L2 | **25.5** | **86.3** | **88.3** | **93.0** | **63.0** |
| MAPOT | **25.5** | **88.4** | **89.3** | **94.8** | **64.5** |
| AVERAGE BOOST | **13.1** | **20.3** | **4.1** | **12.0** | **9.7** |

Table 1: Accuracy of domain adaptation by optimal transport in the latent space of ResNet50 model with the 10 known labels for each class in the target domain on Digits datasets.

| METHOD | MNIST SVHN | SVHN MNIST | MNIST USPS | USPS MNIST | MNIST M-MNIST |
|---|---|---|---|---|---|
| W2GN 10) | 20.4 | **79.9** | **89.1** | **91.2** | **74.1** |
| MM:R (10) | 20.3 | **80.1** | 78.0 | **90.0** | 63.8 |
| W2GN (100) | 20.4 | **79.9** | **89.1** | 91.2 | **74.1** |
| MM:R (100) | 20.3 | **80.2** | 78.0 | **90.2** | 63.8 |
| W2GN (10) | **21.3** | 70.6 | 63.8 | 89.6 | 54.2 |
| MM:R (10) | **21.5** | 75.1 | **79.0** | 85.3 | **70.6** |
| W2GN (100) | **24.4** | 74.4 | 85.1 | **92.0** | 68.5 |
| MM:R (100) | **21.5** | 79.1 | **79.1** | 89.2 | **70.3** |

Table 2: Accuracy of domain adaptation by Neural OT in the latent space of ResNet50 model with the 10 and 100 known labels for each class in the target domain on Digits datasets.

| METHOD | A D | D A | A S | S A | A W | W A | D S | S D | D W | W D | S W | W S |
|---|---|---|---|---|---|---|---|---|---|---|---|---|
| TARGET | 64.8 | 67.9 | 44.5 | 6.2 | 63.3 | 70.3 | 41.6 | 3.0 | 81.6 | 80.8 | 5.5 | 45.6 |
| EMD | 50.7 | 46.2 | 38.4 | 9.3 | 45.2 | 45.6 | 32.7 | 16.4 | 62.6 | 67.1 | 13.6 | 36.7 |
| SINKHORN | 51.1 | 46.3 | 38.0 | 10.1 | 44.7 | 45.5 | 32.9 | 16.5 | 63.5 | 67.1 | 13.1 | 37.2 |
| SINKHORNLPL1 | 51.1 | 46.7 | 38.1 | 10.4 | 45.2 | 45.3 | 33.0 | 16.5 | 63.8 | 68.3 | 13.1 | 37.2 |
| SINKHORNL1L2 | 51.1 | 46.7 | 38.1 | 10.4 | 45.0 | 45.3 | 33.0 | 16.5 | 63.8 | 68.3 | 13.1 | 37.2 |
| MAPT | 45.8 | 48.0 | 37.1 | 11.0 | 38.7 | 47.5 | 36.5 | 4.1 | 60.9 | 61.8 | 6.2 | 39.6 |
| EMD | **70.9** | **72.5** | **56.8** | **29.7** | **64.9** | **73.9** | **56.6** | **47.3** | **75.7** | **75.1** | **40.1** | **60.1** |
| SINKHORN | **70.6** | **72.7** | **57.0** | **31.0** | **65.2** | **73.9** | **56.7** | **47.3** | **77.4** | **74.8** | **39.9** | **60.0** |
| SINKHORNLPL1 | **70.6** | **72.8** | **57.2** | **31.0** | **65.2** | **74.0** | **56.8** | **47.3** | **77.4** | **75.5** | **40.1** | **60.1** |
| SINKHORNL1L2 | **70.6** | **72.8** | **57.2** | **31.0** | **65.2** | **74.0** | **56.8** | **47.3** | **77.4** | **75.5** | **40.1** | **60.1** |
| MAPT | **71.3** | **73.6** | **58.5** | **29.8** | **65.2** | **74.4** | **59.8** | **47.3** | **76.6** | **75.1** | **40.1** | **63.1** |
| AVR BOOST | **29.4** | **35.8** | **33.8** | **66.3** | **32.7** | **38.0** | **41.4** | **70.3** | **18.2** | **11.5** | **70.4** | **38.0** |

Table 3: Results of domain adaptation in the latent space of ResNet50 model on Modern Office-31 dataset in semi-supervised settings with the 10 known labels for each class in the target domain.

## 5.3 RESULTS

Our results show that adversarially based *source fiction* improves the performance of OT. For Digits datasets, our approach achieves improvements on all domains, even on MNIST-SVHN domain adaptation, which is a complicated transformation task. Generally, the simplest EMD method achieved less accuracy and MapT method accuracy slightly higher for all domains; group lasso and Laplacian regularizations for Sinkhorn did not provide notable improvements. The mean and std of our results are less than 10% for all experiments. On average, our method achieves mean improvement equal to 32.7 for all adaptation tasks, with min improvement value equal to 4.1 and max value 70.4. The results with only 3 known labels in class presented in Appendix 6

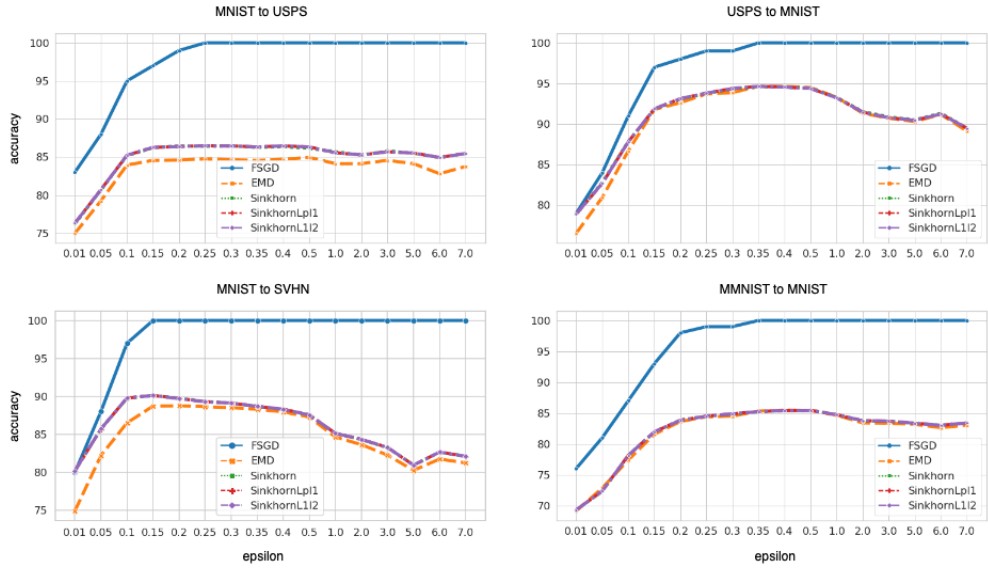

Figure 2: Results of ablation on $\epsilon$ parameter for MNIST to M-MNIST (left) and SVHN to MNIST (right) datasets. FSGD denotes how accurately the source domain model classifies *source fiction* domain obtained with different $\epsilon$ value.

In our opinion, our method advances the performance while OT applications for mass moving assume closeness of source $\Omega_s$ and target $\Omega_t$ distributions (Lee et al., 2019). This assumption does not hold on many empirical datasets, especially in some Digits and Modern Office-31 domains.

While OT maps are cycle monotone, i.e., exhibit a specific structure of the map, thus, transportation $\Omega_t \to \Omega_s$ via OT maps might not apply to some problems, see (Courty et al., 2015, Figure 3) for counter-examples. Also, discrete OT techniques are susceptible to regularization terms (Courty et al., 2015; Dessein et al., 2018; Paty & Cuturi, 2020) and require special scaling (Meng et al., 2021).

As shown in our results, the performance of Neural OT methods is less dependent on the monotonicity of the target domain. It is known that neural networks are data-intensive; using the *source fiction* domain as a target, we have less training data, which makes optimization harder. However, in a case with a higher number of labeled samples in the target domain, source fiction outperforms the standard settings for Neural OT.

While adversarial examples via FSGD are noisy, there appear problems with learning Neural OT on this data. We hypothesize that adversarial training (Goodfellow et al., 2014) for a robust source classifier can be a solution for this problem. Adversarial attack on robust models allows interpolation between classes and the generation of new data points that are close to human perception (Tsipras et al., 2019). It has been shown that this property can be used for image generation, image-to-image translation, in-painting, and super resolution (Santurkar et al.). With robust source classifier *source fiction* domain will contain less noisy samples, which can improve the accuracy of Neural OT methods.

## 5.4 ABLATION STUDY

In this section, we show the adaptation results with different $\epsilon$ values in the FSGD algorithm. We evaluated few transportation tasks: MNIST to USPS, USPS to MNIST, MNIST to M-MNIST, and MNIST to SVHN datasets with different values of $\epsilon$. The value of $\frac{1}{2} \min_{n_1, n_2} \|x_{n_1} - x_{n_2}\|$ is various for different datasets, for SVHN this value is 0.74, for MNIST is 0.29, for M-MNIST is 10.9 and for USPS is equal to 0.85. These values were computed in the latent space of the ResNet50 classifier trained on the corresponding dataset. For each domain, we applied *source fiction* with different values of $\epsilon$ and then fit OT to find a map between target and *source fiction* domain. Our results

show a trade-off between the size of perturbations and the cyclical monotonicity of the resulted domain. See the adaptation accuracy with different $\epsilon$ in Figure 2. It can be noticed that when the $\epsilon$ value becomes larger than $\frac{1}{2}$ min distance between samples, the accuracy of adaptation decreases since the set becomes less monotone. However, with a small value of $\epsilon$, the adaptation achieves the highest accuracy.

## 6 CONCLUSION

We propose an algorithm that modifies domain adaptation toward making target data closer to the domain formed by an adversarial attack. We conducted a range of experiments on different datasets and showed that the OT method works more efficiently in conjugate with our method. In some domains for discrete OT methods, adaptation with source fiction achieved dramatic improvement. The main limitation of our approach is that necessary to have access to the labels in the target domain.

**Multi-domain adaptation:** Usually, in DA, it is necessary to modify classifier architecture and train separately on each target domain (Ganin & Lempitsky, 2015; Long et al., 2018; 2015; 2017; Gretton et al., 2012). OT solves DA by moving target domains samples closer to the source data, and it is unnecessary to change or fine-tune the source classifier. In this paper, we presented an algorithm that improves OT performance, which means that a single source domain classifier can be used to make predictions on a range of target domains more effectively.

**Data privacy:** Nowadays, data are shared on separate devices and usually contain personal information, which is inefficient for data transmission and may violate data privacy. Liang et al. (2020) address a challenging DA setting without access to the source data for higher privacy. In our method, we adapt the source classifier to the new domain without access to the source data, which can also help avoid privacy issues.

**Future work:** Our method admits many straightforward applications:

- Neural OT for adversarial examples generation; fast generation of shareable adversarial examples in a framework similar to Attack-Inspired GAN (Bai et al., 2021) with additional supervising using gradient-based adversarial examples. See experiments using discrete OT for this task in Appendix A.1.

- In our settings, discrete OT methods showed impressive results in domain adaptation with a low number of labeled samples; this ability can be used for low data image recognition, where OT learns to map test dataset to the accurately classifying samples for model learned on train dataset. Such experiments on digits and CIFAR-10 datasets are presented in Appendix A.2

- While OT can solve domain adaptation with target shift and unbalanced classes (Redko et al., 2019; Rakotomamonjy et al., 2020), promising directions is using of *source fiction* for such problems too.

We expect our research to contribute the development of less complicated domain adaptation techniques and open doors for the future application of adversarial attacks cycle monotonicity.

## 7 ETHICS

There is currently no regulatory framework for the use of adversarial attacks in a machine learning environment. Adversarial attacks can be used in real-world scenarios and can harm infrastructures with machine learning models in the pipeline. On the contrary, we examined the adversarial attack property and showed how this technique can be used to improve the performance of machine learning models. We do not provide any new methods for adversarial attacks or protection against them.

## 8 REPRODUCIBILITY

To reproduce our experiment we provide source code in supplementary materials, run `./run_experiments.sh` to start the training process. Details on used hyperparameters are presented in Section 5.2.

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

## A APPENDIX

### A.1 ADVERSARIAL ATTACKS APPROXIMATION

In this section, we propose the usage of OT for Adversarial attack approximation. We used MNIST, MNIST-M, and SVHN datasets and trained three Shirivatsava-like classifiers (Shrivastava et al., 2016) on each dataset individually. To create adversarial examples, we used a simple FSGD method. We transformed 10k images from each training dataset via FSGD into adversarial examples. Having the original and adversarial examples, we used OT to find a map between them. The 10k test samples without associated adversarial examples and labels were input to the OT algorithms too.

| Method | MNIST | USPS | M-MNIST | SVHN | CIFAR-10 |
|---|---|---|---|---|---|
| No attack | 99.0 | 99.0 | 98.0 | 98.0 | 81.0 |
| EMD | 4.0 | 4.0 | 4.0 | 2.0 | 7.0 |
| Sinkhorn | 5.0 | 4.0 | 5.0 | 3.0 | 7.0 |
| Sinkhorn L1Lp | 5.0 | 4.0 | 4.0 | 3.0 | 7.0 |
| Sinkhorn L1L2 | 5.0 | 4.0 | 4.0 | 3.0 | 6.0 |

Table 4: Results of different OT algorithms on digits datasets, on the test set to the adversarial examples transportation task, 10k samples from train set was used to build adversarial examples.

As shown in Table 4, the resulted examples significantly reduce the accuracy of the classifiers. Also, the inference of adversarial examples using OT is much faster than iterative gradient descent methods. Our results confirm that OT can naturally learn the distribution of adversarial examples.

### A.2 IMAGE CLASSIFICATION

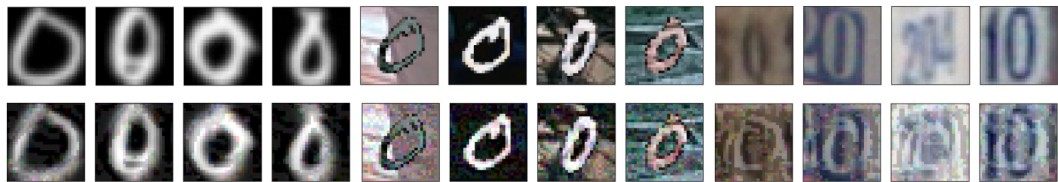

Figure 3: Examples of source fiction in input space for MNIST, USPS, M-MNIST and SVHN.

In comparison to the previous section here, we turn to the opposite side. Our idea is to use *source fiction* for image classification using OT inside the single domain, in other words between test and train data. While discrete OT methods showed impressive results in DA with a low number of labeled samples, we propose to use mapping from the test set to the source fiction dataset of classifier $f(OT(x))$. Source fiction in such a scenario is obtained by the adversarial attack to the model trained on low number training samples. Shirivatsava-like classifiers (Shrivastava et al., 2016) were used for training. We evaluated the Sinkhorn algorithm on MNIST and USPS datasets with 10 known labels and CIFAR-10 with 100 known labels per class. Results are presented in Table 5.

| METHOD | MNIST | USPS | CIFAR-10 |
|---|---|---|---|
| SOURCE | 69.1 | **82.0** | 31.8 |
| SINKHORN | **72.2** | **82.0** | **55.7** |

Table 5: Results of application of Sinkhorn optimal transport in the latent space of classifiers with 10 known labels for each class.

In Figure 3 are presented examples of cycle monotone transformation for *source fiction* via adversarial attack in the dataset space.

### A.3 ADDITIONAL EXPERIMENTS ON DA

In this section, we provide additional experiments using the *source fiction* method:

- Results with 3 (Table 6) and 100 (Table 7) known labels per class in target domain using ResNet50 classifier.
- Digits dataset using Resnet18, with 10 (Table 8) and 100 (Table 9) known labels.
- ResNet18 trained one epoch on source domain Modern Office-31 datasets 10.
- Experiments on the Office-Caltech dataset, unfortunately, full dataset is not available publicly now, so we used an abbreviated version, results are presented in Table 11.
- Resnet18 adaptation on CIFAR10-STL10 adaptation task (Table 12).

| METHOD | MNIST SVHN | SVHN MNIST | MNIST USPS | USPS MNIST | MNIST M-MNIST |
|---|---|---|---|---|---|
| SOURCE | 22.0 | 79.0 | 63.0 | 80.0 | 60.0 |
| EMD | 21.3 | 72.5 | 66.1 | 67.8 | 44.5 |
| SINKHORN | 21.7 | 73.0 | 67.3 | 68.7 | 44.6 |
| SINKHORNLPL1 | 21.7 | 73.4 | 67.3 | 68.8 | 45.0 |
| SINKHORNL1L2 | 21.7 | 73.4 | 67.3 | 68.8 | 45.0 |
| MAPT | 21.8 | 73.4 | 67.4 | 68.8 | 45.0 |
| EMD | **23.11** | **83.5** | **82.6** | **86.5** | **54.7** |
| SINKHORN | **23.7** | **85.0** | **82.6** | **86.8** | **54.8** |
| SINKHORNLPL1 | **23.7** | **85.2** | **86.3** | **86.9** | **54.9** |
| SINKHORNL1L2 | **23.8** | **85.2** | **86.3** | **86.9** | **54.9** |
| MAPT | **23.9** | **85.3** | **86.3** | **86.9** | **55.1** |

Table 6: Accuracy of domain adaptation by optimal transport in the latent space of ResNet50 model with only 3 known labels for each class in the target domain on Digits datasets. The top part of the table represents standard settings the bottom part presents results using *source fiction*.

| METHOD | MNIST SVHN | SVHN MNIST | MNIST USPS | USPS MNIST | MNIST M-MNIST |
|---|---|---|---|---|---|
| SOURCE | 21.68 | 72.54 | 70.0 | 83.3 | 24.3 |
| EMD | 24.24 | 73.5 | 83.1 | 80.1 | 53.6 |
| SINKHORN | 21.6 | 74.5 | 84.0 | 80.4 | 55.6 |
| SINKHORNLPL1 | 21.6 | 74.3 | 85.5 | 81.1 | 55.6 |
| SINKHORNL1L2 | 21.6 | 74.3 | 85.5 | 81.1 | 55.6 |
| MAPT | 21.6 | 74.3 | 85.5 | 81.1 | 55.6 |
| EMD | **25.8** | **85.8** | **86.1** | **91.2** | **52.4** |
| SINKHORN | **27.8** | **85.8** | **86.8** | **91.2** | **56.3** |
| SINKHORNLPL1 | **27.8** | **86.0** | **88.0** | **91.6** | **56.3** |
| SINKHORNL1L2 | **27.8** | **86.0** | **88.0** | **91.6** | **56.3** |
| SINKHORNL1L2 | **27.9** | **86.1** | **88.0** | **91.6** | **56.3** |
| MAPT | **27.9** | **86.1** | **88.0** | **91.6** | **56.3** |

Table 7: Accuracy of domain adaptation by optimal transport in the latent space of ResNet50 model with 100 known labels for each class in the target domain on Digits datasets. The top part of the table represents standard settings the bottom part presents results using *source fiction*.

| METHOD | MNIST SVHN | SVHN MNIST | MNIST USPS | USPS MNIST | MNIST M-MNIST |
|---|---|---|---|---|---|
| SOURCE | 22.0 | 79.0 | 63.0 | 80.0 | 60.0 |
| EMD | 15.4 | 64.3 | 77.0 | 80.8 | 70.8 |
| SINKHORN | 16.0 | 65.2 | 77.9 | 81.2 | 70.9 |
| SINKHORN L1LP | 16.8 | 65.7 | 79.8 | 85.1 | 71.8 |
| SINKHORN L1L2 | 16.0 | 65.7 | 79.8 | 85.1 | 71.8 |
| MAPT | 16.1 | 67.1 | 79.8 | 86.2 | 71.8 |
| EMD | **35.1** | **87.1** | **85.2** | **95.2** | **82.3** |
| SINKHORN | **38.0** | **88.3** | **88.3** | **95.2** | **83.5** |
| SINKHORN L1LP | **37.1** | **88.3** | **88.2** | **95.2** | **83.5** |
| SINKHORN L1L2 | **37.1** | **88.3** | **88.3** | **95.2** | **83.5** |
| MAPT | **38.0** | **90.0** | **88.2** | **95.2** | **83.5** |

Table 8: Accuracy of domain adaptation by optimal transport in the latent space of ResNet18 model with the 10 known labels for each class in the target domain on Digits datasets. The top part of the table represents standard settings the bottom part presents results using *source fiction*.

| METHOD | MNIST SVHN | SVHN MNIST | MNIST USPS | USPS MNIST | MNIST M-MNIST |
|---|---|---|---|---|---|
| SOURCE | 22.0 | 79.0 | 63.0 | 80.0 | 60.0 |
| EMD | 18.4 | 71.3 | 85.0 | 88.0 | 75.1 |
| SINKHORN | 20.0 | 76.2 | 85.3 | 86.0 | 81.9 |
| SINKHORN L1LP | 20.8 | 76.5 | 85.2 | 85.0 | 81.7 |
| SINKHORN L1L2 | 20.0 | 75.5 | 85.2 | 86.0 | 81.7 |
| MAPT | 20.9 | 76.2 | 85.0 | 86.9 | 81.6 |
| EMD | **24.3** | **89.1** | **87.3** | **96.0** | **86.3** |
| SINKHORN | **27.1** | **90.0** | **89.3** | **96.0** | **87.1** |
| SINKHORN L1LP | **28.3** | **90.0** | **89.2** | **96.0** | **87.1** |
| SINKHORN L1L2 | **28.3** | **90.0** | **89.2** | **96.0** | **88.1** |
| MAPT | **29.0** | **90.2** | **89.3** | **96.2** | **88.0** |

Table 9: Accuracy of domain adaptation by optimal transport in the latent space of ResNet18 model with the 100 known labels for each class in the target domain on Digits datasets. The top part of the table represents standard settings the bottom part presents results using *source fiction*.

| METHOD | A D | D A | A S | S A | A W | W A | D S | S D | D W | S W | W S |
|---|---|---|---|---|---|---|---|---|---|---|---|
| SOURCE | 37 | 43 | 25 | 8 | 57 | 41 | 20 | 2 | 65 | 3 | 27 |
| EMD | 57 | 42 | 27 | 16 | 53 | 38 | 32 | 28 | 53 | 19 | 37 |
| SINKHORN | 58 | 43 | 27 | 17 | 53 | 40 | 32 | 29 | 54 | 20 | 38 |
| SINKHORNLpL1 | 58 | 43 | 27 | 17 | 53 | 40 | 32 | 29 | 53 | 20 | 38 |
| SINKHORNL1L2 | 58 | 43 | 27 | 17 | 53 | 40 | 32 | 29 | 53 | 20 | 38 |
| MAPT | 58 | 43 | 27 | 17 | 53 | 40 | 32 | 29 | 53 | 20 | 38 |
| EMD | **76** | **67** | **51** | **44** | **73** | **63** | **59** | **60** | **76** | **45** | **61** |
| SINKHORN | **77** | **67** | **51** | **43** | **75** | **63** | **59** | **57** | **77** | **41** | **60** |
| SINKHORNLpL1 | **77** | **67** | **51** | **43** | **75** | **63** | **59** | **57** | **77** | **41** | **60** |
| SINKHORNL1L2 | **77** | **67** | **51** | **43** | **75** | **63** | **59** | **57** | **77** | **41** | **60** |
| MAPT | **77** | **67** | **51** | **43** | **75** | **63** | **59** | **57** | **77** | **41** | **60** |
| AVERAGE BOOST | **25** | **37** | **47** | **63** | **27** | **39** | **45** | **53** | **30** | **57** | **39** |

Table 10: Results of domain adaptation in the latent space of ResNet18 model on Modern Office-31 dataset in semi-supervised settings with the 10 known labels for each class in the target domain

## A.4 ENERGY-BASED MODELS FOR EPSILON FREE SOURCE FICTION

As shown in section 5, our algorithm depends on the size of perturbations, to avoid it and make our method $\epsilon$ free, we propose to use an input convex energy-based classifier. Energy-based learning provides a unified framework for many probabilistic and non-probabilistic approaches, particularly for non-probabilistic training of graphical models, including discriminative and generative

| METHOD | A W | W A | A C | A D | D A | D C | D W | D C |
|---|---|---|---|---|---|---|---|---|
| SOURCE | 49 | 27 | 65 | 70 | 42 | 26 | 62 | 36 |
| EMD | 42 | 44 | 30 | 59 | 40 | 24 | 53 | 38 |
| SINKHORN | 43 | 43 | 30 | 60 | 37 | 24 | 52 | 38 |
| SINKHORNLPL1 | 42 | 43 | 30 | 60 | 37 | 25 | 53 | 38 |
| SINKHORNL1L2 | 42 | 43 | 30 | 60 | 37 | 25 | 53 | 38 |
| MAPT | 42 | 43 | 30 | 60 | 37 | 25 | 53 | 38 |
| EMD | **54** | **51** | **52** | **68** | **67** | **39** | **69** | **47** |
| SINKHORN | **54** | **51** | **52** | **68** | **67** | **39** | **69** | **48** |
| SINKHORNLPL1 | **54** | **51** | **52** | **68** | **67** | **39** | **69** | **48** |
| SINKHORNL1L2 | **54** | **51** | **52** | **68** | **67** | **39** | **69** | **48** |
| MAPT | **54** | **51** | **52** | **68** | **67** | **39** | **69** | **48** |
| AVERAGE BOOST | 21 | 15 | 42 | 12 | 3 | 36 | 23 | 20 |

Table 11: Results of domain adaptation in the latent space of ResNet18 model on Office-31-Caltech dataset in semi-supervised settings with the 10 known labels for each class in the target domain

| METHOD | SOURCE | EMD | SINKH | SINKH L1LP | SINKH L1L2 | MAPT |
|---|---|---|---|---|---|---|
| CIFAR→STL | 37.0 | 48.1 | 48.1 | 48.0 | 48.0 | 48.1 |
| CIFAR→SF | | **51.1** | **51.0** | **51.0** | **51.0** | **51.0** |
| STL→CIFAR | 75.0 | 74.1 | 74.1 | 74.1 | 74.1 | 74.1 |
| STL→SF | | **76.2** | **76.2** | **76.2** | **76.2** | **76.2** |

Table 12: Results on MNIST and USPS dataset in semi-supervised settings. U is USPS, M is MNIST, SF is source fiction. The top table presented results for the settings with the 10 known labels for each class in the target domain, and the bottom table presents the result with the 100 known labels for each class

.

approaches and conditional random fields, graph-transformer networks, maximum margin Markov networks, and several manifold learning methods (LeCun et al., 2006). In energy-based settings for some given fixed $x$ and possibly some fixed elements of $y$ we can perform inference by:

Energy-based learning approaches can be considered as an alternative to probabilistic estimation for prediction, classification, or decision-making tasks. The energy-based representation must capture both the discriminative interactions between $x$ and $y$ and allow for efficient combinatorial optimization over $y$.

### A.4.1 CONVEX ENERGY-BASED INFERENCE

Here is our approach in more detail. First of all, we build a model $f_\theta$ that is Partly Input Convex Neural Network (PICNNs) (Amos et al., 2017) over $x$ instead of $y$ (Amos et al., 2017), it is made possible to apply convex inference over $x$. Secondary we train our model $f_\theta$ on the source domain $(X_s, Y_s)$ in setup equal to Structured Prediction Energy Networks (SPEN) (Belanger & McCallum, 2016), but instead of multi-label classification, we simply perform multi-class classification.

In the next step, we apply the core idea of our approach, we use a model trained on $(X_s, Y_s)$, to create a *source fiction* domain which is a cyclical monotone to the target domain $(X_t, Y_t)$ and contain features from the source domain samples. We simply input $\mathbf{x}_i^t$ samples to the model $f_\theta$ and apply inference over it with the fixed $\mathbf{y}_i^t$:

$$\arg\min_x f(x, y; \theta) \tag{13}$$

As stated before, according to Rockfellar (Rockafellar, 1966) and Breirer's theorems (Theorem 1.22 (Santambrogio, 2015)), we know that the gradient of a convex function is cyclical monotone and solves a Monge problem (Villani, 2008).

We used the gradient descent method to inference and while our $f_\theta$ is input convex over the $x$, applying this procedure over the target samples we can collect a new source fiction domain $(X_f, Y_f)$ where $X_f$ samples are class-wise cycle monotone to $X_t$. And according to the theorems presented before, there exists optimal transport that can solve transportation from $X_t$ to $X_e$ inside each class. Connecting resulted domain with penalty $\Omega_{semi}(\gamma) = \langle \gamma, \mathbf{M} \rangle$ by a $n_s \times n_t$ cost matrix $M \rangle$ where $M(i, j) = 0$ when $y_i^s = y_j^t$ and $+\infty$ otherwise (Courty et al., 2016; Yan et al., 2018) we achieve improvement in accuracy without $\epsilon$ value to restrict the seize of perturbations.

### A.4.2 EXPERIMENTS

We tested our model on MNIST (LeCun & Cortes, 2010) and USPS datasets (Hull, 1994). In both experiments, we train PICNNs with a one hidden layer size of 100, in the SPEN settings using SGD (Ruder, 2016) optimizer with a learning rate equal to 1e-3 and momentum equal to 0.9.

| METHOD | SOURCE | EMD | SINKH | SINKH L1LP | SINKH L1L2 | MAPT |
|---|---|---|---|---|---|---|
| U→M | 28.84 | 39.33 | 37.76 | 31.14 | 11.8 | 28.15 |
| U→SF | - | **76.13** | **76.13** | **76.13** | **76.13** | **68.21** |
| M→U | 28.56 | 34.43 | 30.42 | 25.75 | 14.29 | 28.68 |
| M→SF | - | **68.72** | **62.74** | **57.92** | **16.02** | **56.11** |
| | | | | | | |
| U→M | 28.84 | 45.54 | 35.52 | 29.19 | 9.41 | 33.58 |
| U→SF | - | **86.39** | **86.39** | **86.34** | **86.34** | **72.09** |
| M→U | 28.56 | 45.01 | 29.9 | 26.09 | 9.68 | 48.12 |
| M→SF | - | **85.80** | **77.60** | **69.31** | **12.59** | **61.35** |

Table 13: Results on MNIST and USPS dataset in semi-supervised settings. U is USPS, M is MNIST, SF is source fiction. The top table presented results for the settings with the 10 known labels for each class in the target domain, and the bottom table presents the result with the 100 known labels for each class

.

Based on the POT library (Flamary et al., 2021) we tested the different variations of discrete optimal transport in this task. First of all, we tested and basic EMD and Sinkhorn (Sinkh) (Cuturi, 2013) algorithms. Then we tested regularized versions of the Sinkhorn algorithm with a group lasso regularization (L1L2) and Laplacian regularization (L1LP) (Courty et al., 2015). Finally, our method is benchmarked on MapT (Perrot et al., 2016).

The results presented in Table 13. In the table, we can see that mapping to the source fiction domain improves the accuracy of the optimal transport algorithms, but due to inference problems with energy-based models, accuracy is less than in settings with standard networks.

