# OpenReview forum: "Cycle monotonicity of adversarial attacks for optimal domain adaptation"
_ICLR.cc/2022/Conference — ICLR 2022 Submitted_

### Official Review · Reviewer_Kkzh · 2021-10-28

**Correctness:** 3
**Technical Novelty And Significance:** 2
**Empirical Novelty And Significance:** 2
**Recommendation:** 3
**Confidence:** 4

**Main Review:**

Strengths:
- Studying the use of adversarial attacks for domain adaptation is an interesting topic, which has not been extensively covered yet.
- The results using discrete OT methods suggests that the proposed approach can bring some improvement.


Weaknesses:

1) Motivation/intuition behind the proposed method.
1.1) The authors spend time explaining the notion of cycle monotonicity and showing that FSGD attacks are cycle monotone. However, they do not explain why this is important, what is the connection between cycle monotonicity of adversarial attacks and the success of domain adaptation. This explanation is needed, as the current version of the paper gave me no intuition as to why the method should work.

1.2) In fact, the method is quite counter-intuitive. Let me describe Algorithm 1 in words: Given a classifier trained on the source data, one generates adversarial examples for the labeled target examples. As such, these adversarial examples will be misclassified by the source classifier. One then maps the vanilla target examples to the domain of the adversarial examples, and applies the same source classifier. As this classifier is ineffective for the adversarial examples, I do not see why it would work for the transformed unlabeled target samples. I would highly appreciate if the authors could explain this.

2) The experimental results are not truly convincing.
2.1) While the results with discrete OT are fine, the ones with neural OT (Table 2) are disappointing. In short, there is no clear benefit of the method in this case. This is problematic because neural OT methods tend to give better results than discrete OT ones, as can be seen by comparing Table 1 and Table 2. This significantly reduces the attractiveness of the method.

2.2) The results of neural OT are only shown on the Digits datasets, not the Modern Office-31 one.

2.3) As there are no references, is not entirely clear to me if the discrete OT and neural OT methods used in the experiments correspond to the state-of-the-art methods for OT-based domain adaptation. If not, a comparison to such methods would be needed.

3) Clarity: Although the paper is altogether clear enough, it could be improved in several ways.
3.1) The introduction would not be easy to follow for someone not familiar with the notion of cycle monotonicity. It would help if the authors could give an intuitive explanation of what it is and why it is useful in this context.

3.2) In the second paragraph, the authors state that the conventional OT approach for domain adaptation is to transform the target domain to the source one. I would argue that the reverse transformation is more common (e.g., Rakotomamonjy et al., 2020, Damodaran et al., 2018).

3.3) The background on OT is quite detailed, which may not be truly necessary. This could be moved to the appendix.

3.4) By contrast, the cycle monotonicity equation (eq. 7) lacks some information, i.e., the fact that $y_{N+1} = y_{1}$. Note also that, above the equation, it should be $x_N$ and $y_N$ instead of $x_i$ and $y_i$.

3.5) The result in Lemma 1 is not surprising considering the assumption on $\epsilon$. As it is intuitive, the authors could consider moving the proof to the appendix.

3.6) Minor comment: The use of $N$ in Algorithm 1 is a bit unfortunate, as it typically refers to the number of samples.


**Summary Of The Paper:**

This paper studies the use of adversarial attacks and optimal transport (OT) in the context of semi-supervised domain adaptation. The authors show that adversarial attacks satisfy the cycle monotonicity property. They then propose an algorithm that generates adversarial examples for the labeled target samples using the source classifier, and maps the vanilla target samples to this new domain via OT.


**Summary Of The Review:**

Although this paper studies an interesting scenario, I am convinced neither by the motivation behind the proposed approach, nor by the experimental results.

---

> ### Author Response · Authors · 2021-11-14
> **Answer to Reviewer Kkzh**
>
> Thank you for your great suggestions on paper improvement. Please find below the answers to your questions.
>
> $\textbf{(Q1) In fact, the method is quite counter-intuitive. I do not see why it would work for the transformed unlabeled target samples.}$
>
> We think that you might have misunderstood our algorithm.  In our method, we use "anti" adversarial attack, see Section 4. We use the vulnerability of classifiers to adversarial attacks to generate correctly classified samples and then transport the target samples into such samples.
>
> $\textbf{(Q2) Neural OT methods tend to give better results than discrete OT ones... The results of neural OT are only shown on the Digits datasets.}$
>
> Training Neural OT takes hours while DA with Discrete OT takes a few minutes. We find that Neural OT methods might be unstable and hard to train; the reasons why our method works worse for neural OT methods and ways to avoid this problem are presented in the Results section. Benchmarking the Digits datasets, we find that our method with Discrete OT outperforms Neural OT, so we decided to skip the experiments with Neural OT on Modern Office datasets. Neural OT is a great technique applicable in other tasks than DA.
>
> $\textbf{(Q3) It is not entirely clear to me if the discrete OT and neural OT methods correspond to the state-of-the-art methods.}$
>
> Current methods are fundamental, and to the best of our knowledge, they are SOTA for unsupervised and supervised DA settings[1]. We tested our method on a range of fundamental OT algorithms to show that our method can work on different OT solvers that involve cycle monotonicity. There are many different OT algorithms for other types of DA tasks (Multi-source DA, DA with target shift, etc.). They are out of the scope of this paper.
>
>
> References
>
> [1] R. Flamary et al.  Pot:Python optimal transport. Journal of Machine Learning Research, 22(78): 1–8, 2021.

---

> > ### Comment · Reviewer_Kkzh · 2021-11-24
> > **Post-rebuttal feedback**
> >
> > Thank you for your answers.
> >
> > Regarding Q1, I had indeed misunderstood what was referred to as target for the attack. So I acknowledge that the method is not as counter-intuitive as I originally thought.
> >
> > I nonetheless believe that the motivation behind the method and the reason why it works should be better explained in the paper, and I remain somewhat unconvinced by the experimental results. As such, I would recommend the authors to thoroughly revise their manuscript and resubmit it to a future venue.

---

### Official Review · Reviewer_kk9C · 2021-10-30

**Correctness:** 3
**Technical Novelty And Significance:** 3
**Empirical Novelty And Significance:** 3
**Recommendation:** 3
**Confidence:** 4

**Main Review:**

Strengths: this paper aims to use adversarial attacks to generate good samples for the task of semi-supervised domain adaptation. It takes three steps: first, pretrain a source classifier $h$; second, attack the source classifier with labeled target samples; finally, use optimal transport to align the unlabeled target samples and label them using $h$. This approach seems interesting. I appreciate the authors' effort to conduct experiments on various vision datasets and several OT algorithms. The new approach seems to perform better than unsupervised domain adaptation.

Weaknesses:

1. cyclical monotonicity is equivalent to optimal transport (Exercise 2.21, [1]). The adversarial attack in this paper essentially says that after small perturbations, the new samples are still the optimal plan when transporting the initial samples. This is always true if the perturbations are small enough, and this paper gives the upper bound of the perturbation. While cyclical monotonicity is a good evaluation metric to check optimal transport. I feel it is unnecessary to introduce this concept in this work, and doing so would intimidate and confuse the readers. All we need to say is that the small perturbations are not far so that the modified samples are still the closest to the original samples, among all permutations.

2. The definition of cyclical monotonicity (CM) is not clear in this paper. CM would require a subset $\Gamma$ such that for all $(x_1, y_1), \cdots, (x_m, y_m) \in \Gamma$, eq. (7) is satisfied. However, it is not clearly written before eq. (7) and Lemma 1. Moreover, in the paper "cycle monotonicity" is used several times, instead of "cyclical monotonicity" [1].

3. On the experimental side, I am a bit worried that the comparison is not fair. The authors compared the "standard setting" with their semi-supervised settings. This "standard setting" is not detailed in the paper. I can guess from Figure 1 that this standard setting is unsupervised domain adaptation (Gaini et al JMLR 2016, Domain-Adversarial Training of Neural Networks), but in Algorithm 1, labeled data from target domain is used. Thus it is not fair to compare unsupervised domain adaptation with semi-supervised domain adaptation, as in the latter case we have additional labeled data from the target.

4. From Figure 2 it seems that with larger perturbation, the source classifier can perform better on the source fiction domain. Could the authors explain why it happens?

5. How many labeled samples have you used in the target domain? If it is large then there is no need to do domain adaptation: we can just train on the target domain. I think we could see it more clearly if we train a new classifier on both the source domain and the labeled part of the target domain, and compare with Algorithm 1. Moreover, adversarial training methods should be used so as to achieve better generalization property [2].


[1] Cédric Villani, Topics in Optimal Transportation, American Mathematical Soc., 2003.

[2] Song et al, IMPROVING THE GENERALIZATION OF ADVERSARIAL TRAINING WITH DOMAIN ADAPTATION, ICLR 2019.


**Summary Of The Paper:**

This paper proposes using the method of adversarial attacks for the task of semi-supervised domain adaptation. The adversarial perturbations are constrained to satisfy cyclical monotonicity [1].

 [1] Cédric Villani, Topics in Optimal Transportation.

**Summary Of The Review:**

Using adversarial attacks for domain adaptation is an interesting idea and has been explored before (https://arxiv.org/pdf/1810.00740.pdf). This paper proposes generating adversarial attacks and using OT to label the target domain. However, I think the current version is not well written: the theory can be greatly simplified and the experimental settings are not clear. Therefore I would recommend rejection.

---

> ### Author Response · Authors · 2021-11-14
> **Answer to Reviewer kk9C**
>
> Thank you for your great suggestions on paper improvement. Please find below the answers to your questions.
>
> $\textbf{(Q1) "standard setting" is not detailed in the paper}$
>
> Sorry for this confusion; we will insert more details about it into the paper. The standard settings are not unsupervised settings but standard settings for semi-supervised OT between source and target domains[1].
>
> $\textbf{(Q2) From Figure 2, it seems that with larger perturbation... It’s still happening because of distance between samples.}$
>
> With larger perturbations, the prediction of the classier on perturbed samples becomes more accurate (see FSGD curve in Figure 2) until perturbation size achieves the proposed bound. With perturbations a little bit larger than the proposed bound, most of the samples in source fiction are still CM with respect to the target, and the method still works.
>
> $\textbf{(Q3) How many labeled samples have you used in the target domain?}$
>
> Appendix sections presented the adaptation results with only 3 labels per class, with accuracy very similar to the case with 10 and 100 labeled samples per class. We will include these experiments in the main part of the paper.
>
> References
>
> [1] Nicolas Courty, Re ́mi Flamary, Devis Tuia, and Alain Rakotomamonjy. Optimal transport for do main adaptation. CoRR, abs/1507.00504, 2015

---

> > ### Comment · Reviewer_kk9C · 2021-11-24
> > **Thank you for your response**
> >
> > I would like to thank the authors for the reply and clarifying my concerns. However, my concerns on the presentation remain after the rebuttal and currently I cannot recommend acceptance. I would anticipate the following in a revised version.
> > - a clearer writing
> > -  more comparisons with semi-supervised DA methods
> > -  clarification of the difference with un-supervised DA methods and comparison

---

### Official Review · Reviewer_Ux4r · 2021-11-01

**Correctness:** 3
**Technical Novelty And Significance:** 2
**Empirical Novelty And Significance:** 2
**Recommendation:** 3
**Confidence:** 4

**Main Review:**

## Strengths:
1. There is some setting where the proposed algorithm improves over the baseline.
2. From a non-expert of the field perspective, the experimental results seem relatively extensive.

## Weaknesses:
1. The theory is relatively weak.
It is fine to prove relatively weak results but it seems to me that a lot of space is used to only apply the triangle inequality. Maybe this space should be used to show to what extent the assumption of Lemma 1 holds or examples of assumption of the data distribution that insure that such an assumption holds ($\min_{n_1,n_2} \|x_{n_1} - x_{n_2}\| \geq 2\epsilon$).
2. The explanation of why this method works could be developed more.
It is said that this method works because the data points in $\Sigma_f$ contain the non-robust features of the source dataset “Such attack adds to the image features of the class to which the image really belongs (Ilyas et al., 2019).“
This reason is not investigated experimentally. For instance, these reasons should have two consequences:
A. The predictions should be non-robust (i.e. very sensitive to adversarial perturbation)
B. such a technique should not work with a robust classifier $f_\theta$.

3. The writing of the paper could be significantly improved. Its clarity and quality (in terms of writing is significantly under ICLR standard)
In my opinion, this paper is not polished enough for a venue such as ICLR. There are many small mistakes in the paper. I also found the flow of this paper confusing:
- Why does cycle monotonicity suggests using optimal transport?
- Is the OT map between $\Sigma_f$ and $\Sigma_t$ or $\Sigma_{semi}$ and $\Sigma_f$? It it seems that is has to be between $\Sigma_f$ and $\Sigma_t$ (in order to classify non-labeled examples) but Algorithm 1 mentions  $\Sigma_{semi}$ and $\Sigma_f$.
- Usually, the transport map is from the source to the target domain (see Courty et al 2016). Why is it the other way in this work?
- The conclusion contains unfinished sentences “The main limitation of our approach is that necessary to have access to the labels in the target domain.“

4. There is no rigorous measure of uncertainty in the experiments:
“The mean and std of our results are less than 10% for all experiments.“ What are you referring to?
If the stds are less than 10% it means that you should not bold any results that are within error bars.
Usually, error bars are 1.96 stds. Thus in order to have a significant difference between the baseline and your results you may need a difference of 1.96 * 2 * 10= 39.2 %
I am confident that you can reduce these confidence intervals with more rigorous experiments but with the current uncertainties, the results displayed in this paper may not be significant.

### Minor comment:
- The $p$-Wasserstein distance is defined for $p \geq 1$
- “Are used to penalty”
- Cost matrix $M \rangle$
- Lemma 1: we need to assume that $n_1 \neq n_2$
- What does $\bar{1,N}$ mean?
- Algorithm 1 $X_f$ should be $\Omega_f$
- Steps $n,...,N$
- $x’_{n+1}$
- “*The* source domain classifier”
- “Such *an* an attack”


**Summary Of The Paper:**

This paper proposes a technique for domain adaptation in the semi-supervised setting (we have access to some labels of the target domain). The idea developed in the paper is to train a classifier (source classifier) on the source domain in order to use the latent representations of this classifier for two things:
1. “anti” Adversarial examples are computed on the labeled target data in order to have them classified correctly by the source classifier. (I call them anti adversarial examples since the goal is to change the labeled target example to improve the accuracy of the source classifier). Such a modified set is called $\Omega_f$
2. An Optimal transport algorithm (OT) is used to transport the latent representations of $\Omega_t$ to the latent representations of the target domain $\Omega_f$ (with a consistency constraint between labeled examples of $\Omega_t$ and $\Omega_f$)

The authors eventually try their algorithm experimentally.


**Summary Of The Review:**

As a summary the ideas from this paper are interesting but it is clearly an unfinished work. I encourage the authors to polish their paper and improve their experimental results to eventually resubmit their work.

---

> ### Author Response · Authors · 2021-11-14
> **Answer to Reviewer Ux4r**
>
> Thank you for your valuable feedback. Please find below the answers to your questions.
>
> $\textbf{(Q1) The predictions should be non-robust. Such a technique should not work with a robust classifier? }$
>
> In our paper, we apply perturbations inside the class, i.e., making the sample look like the most representative sample from this class, as you named it: "anti" adversarial examples. In practice, it is easier to perturb an image to look more like the true class sample because the sample already consists of some features from its real class samples, and perturbations larger than the proposed bound are not necessary to create source fiction for the robust networks.
>
> Even if we suppose the scenario where we need a robust predictor to large perturbations, our method can still be used to solve domain adaptation. We can use our method to predict labels to target samples by source classifier and then train the new robust classifier on the target samples labeled by our method.
>
> $\textbf{(Q2) Why does cycle monotonicity suggests using OT?}$
>
> Please find the answer to this question in answer to the shared questions in section 1.2
>
> $\textbf{(Q3) Usually, the transport map is from the source to the target domain (see Courty et al 2016).
> Why is it the other way in this work?}$
>
> In (Courty et al 2016) settings after the transport from the source to the target domain, it is necessary to train a classifier on labeled target samples to estimate the accuracy of DA. In our paper, we trained one classifier on the source domain and then mapped all related target datasets to the source fiction domain. It's not necessary to train a separate predictor on each target to estimate the quality of the map. Target to the source notation is not novel and commonly used in DA [1]
>
> $\textbf{(Q4) There is no rigorous measure of uncertainty in the experiments}$
>
> Thank you for your remark. We will include the exact values of stds to the paper.
>
>
> References
>
> [1] Mei Wang, Weihong Deng. Deep Visual Domain Adaptation: A Survey. https://arxiv.org/abs/1802.03601

---

> > ### Comment · Reviewer_Ux4r · 2021-11-24
> > **Thank you for your asnwers**
> >
> > Thank you for your answers.
> > I have read your answer and the other reviews. The author did not submit any revision of the paper.
> > I maintain my opinion that this work is promising but is rather unfinished and thus maintain my score. I encourage the author to resubmit their work.

---

### Official Review · Reviewer_bUjz · 2021-11-03

**Correctness:** 3
**Technical Novelty And Significance:** 3
**Empirical Novelty And Significance:** 2
**Recommendation:** 5
**Confidence:** 4

**Main Review:**

**Strengths:**
- The paper develops an interesting connection between "slightly perturbed samples" (possibly misnamed adversarial examples) that gives a cycle monotone map with respect to the original samples.

- Empirical results demonstrate that the proposed algorithm can boost the performance of prior OT-based domain adaptation algorithms.

- The paper gives some interesting future direction ideas including the fact that the source dataset is not needed, rather just the source classifier.


**Weaknesses:**
- If my understanding is correct, the discrete OT versions only work if you have access to the full target dataset *at test time* since you must learn a map from $\Omega_{semi}$ to $\Omega_f$ to do the target classification.  Thus, this mapping step must be done whenever a new sample comes in---i.e., it cannot be done once during training and then work for any new test sample.  Other methods like the neural-based methods should work for any new test point after training. However, the results in the case of neural-based approaches are significantly weaker.

- Fundamentally, this paper only compares to OT-based domain adaptation methods.  What about adversarial-learning based domain adaptation.  It is unclear if this would actually produce state-of-the-art results.  Are there some baseline semi-supervised domain adaptation methods that could be compared against (or even unsupervised domain adaptation methods) to compare against?  It is unclear that OT-based methods are in fact state-of-the-art currently. If they are not, then what is the reason to use OT-based domain adaptation over the state-of-the-art methods (i.e., what additional benefit is gained by using OT if performance is the main concern)?

- The paper writing, theoretical development, and algorithm descriptions could be clarified.  There are typos throughout the paper (see below).  For example, more intuition about the lemma and the corollary are important.  What does this mean intuitively?  Does it mean that the "pair" of any sample is closest to it's corresponding adversarial example?  More generally, does this mean that the adversarial perturbations *implicitly* give the *optimal* mapping between samples from the data distribution and samples from an "adversarial distribution"?  Relatedly, Given that cycle monotonicity is a key term used, there should be more background and a more precise definition in the background section, rather than just an informal statement.

**Other comments or questions**
- By restricting the perturbation epsilon, can the perturbations be thought as finding nearby parts of the classifier that classify correctly?  Or put another way, does this find samples that are closer to the "correct" parts of the source classifier?

- Are these actually the opposite of adversarial examples because you want to decrease the loss function for the "perturbed sample"?  It seems that they are "adversarial" only in the fact that they are small perturbations but they are NOT adversarial in terms of trying to fool the source classifier.  If this is true, then I think the terms should be different.  Something like "perturbed" samples rather than "adversarial" samples.  Also, in Corollary 1, this shouldn't necessarily be called an "attack" since it is not trying to fool a classifier.

- What is $\overline{1,N}$ in Lemma 1?

- There are multiple typos and small issues with wording (e.g., "FSGD" which is probably meant to be "FGSM").  For example the sentence "While OT maps are cycle monotone, i.e., exhibit a specific structure of the map, thus, transportation $\Omega_t \to \Omega_s$ via OT maps might not apply to some problems, see (Courty et al., 2015, Figure 3) for counter-examples." doesn't make sense.  Are you saying that the cycle monotonicity causes OT maps to be poor, or are you saying that they could be poor even though they are cycle monotonic?

- For clarity, it should be more strongly emphasized that you are applying OT methods in the latent space of the classifier, NOT the raw pixel space.

- The algorithm and figure should be presented after they are reference in section 4.

- Table captions should be above tables.


**Summary Of The Paper:**

The paper proposes an algorithmic trick (related to adversarial examples) to enhance existing semi-supervised domain adaptation techniques.
The enhanced method does require at least a few labeled samples in the target domain.
The paper first shows that if you perturb target samples by a small enough epsilon, you can maintain cycle monotonicity (and thus it is strongly related to the optimal mapping between two empirical distributions by construction).
The paper proposes to create a "source fiction" dataset where each labeled target point is perturbed by a bounded epsilon amount (as in adversarial examples) but *unlike* adversarial examples, you are trying to move the point to be **correctly** classified.
Then, all the target data is mapped to this "source fiction" dataset and the classifier is applied to this mapped target data.
The paper shows that this idea can possibly boost the performance of existing OT-based domain adaptation methods.


**Summary Of The Review:**

Overall, the connection between perturbed samples and cycle monotone maps is interesting. However, the paper lacks clarity and intuition about why this is good or useful.  From the experimental side, the paper does not compare to any non-OT domain adaptation methods and is challenging to apply in real-world scenarios because either an OT problem would need to be solved for each new target sample (that wasn't in the original target dataset) or a neural OT would need to be used (whose results were less significant).

---

> ### Author Response · Authors · 2021-11-14
> **Answer to Reviewer bUjz**
>
> Thank you for your valuable feedback and great suggestions on paper improvements. Please find below the answers to your questions.
>
> $\textbf{(Q1) The discrete OT versions only work if you have access to the full target dataset}$
>
> Thank you very much for this important note. To solve this problem, we can use our method to predict labels to target samples by source classifier and then train the new classifier or fine-tune source classifier on the target samples labeled in this way. We made these experiments, and the target classifier achieves the same accuracy as reported in our tables. We will include these experiments in Appendix.
>
> $\textbf{(Q2)  This paper only compares to OT-based DA methods.}$
>
> Discrete OT methods are a simple and fast DA technique in contrast with adversarial-based and other neural-based approaches. OT-based methods like EMD, Sinkhorn, and MapT take a few minutes to apply DA for datasets of moderate sizes. Importantly, OT methods applied for DA have theoretical guarantees [1]. The adversarial approaches for DA are more complex, typically solve the challenging mini-max optimization, and take hours of training.
>
> $\textbf{(Q3) By restricting the perturbation epsilon, can the perturbations be thought as finding nearby parts of the classifier that classify correctly?}$
>
> Yes, we can consider the perturbed samples as samples that are closer to the "correct" parts of the source classifier. Our method achieves the high accuracy of adaptation even if, after training on the source domain, the classifier achieves low accuracy because, on the correct parts, its accuracy is close to 100$\%$.
>
> $\textbf{(Q4) Are you saying that the cycle monotonicity causes OT maps to be poor?}$
>
> The statement here is that OT tries to find a cycle-monotone map between domains. The adaptation results can be poor in the standard OT for DA settings because the cycle-monotone map may doesn't save the class labels, which is very important during DA.
>
> References
>
> [1] Redko, I., Habrard, A., Sebban, M.: Theoretical analysis of DA with OT. In: ECML/PKDD. (2017) 737–753

---

> > ### Comment · Reviewer_bUjz · 2021-11-24
> > **Post-rebuttal response**
> >
> > Thank you for your concise answers to a few questions. I am still unconvinced that the paper ideas are well-formulated.  While I appreciated the basic idea from Q1, this should possibly be integrated more fully into the whole flow.  Also, I still question using discrete OT methods for high-dimensional data and the experimental results are unconvincing given the lack of comparisons.  If the paper had deeper theory and argued for the key benefits discrete OT over adversarial/NN approaches, then it might be okay to not compare.  But as it is, the theory and insights are vague and thus, if the paper rests on empirical results, it will need comparisons to adversarial/NN approaches as they have been shown successful in part at least for this task.  Thus, I am unable to change my recommendation and score.

---

### Author Response · Authors · 2021-11-14
**Answers to the shared questions**

We thank the reviewers for their insightful comments. We are encouraged that you discovered that the connection between perturbed samples and cycle monotone maps is interesting. Please find below the answers to the shared questions.

$\textbf{1.1 Motivation behind the proposed approach}$

Optimal transport (OT), more precisely Discrete OT, is probably the simplest method that you can use to apply domain adaptation (DA). In DA, OT methods try to find a cyclical monotone (CM)[1] map between domains $\Omega_t$ and $\Omega_s$. However, this map is not guaranteed to move samples from $\Omega_t$ to the same class in the domain $\Omega_s$.

Our goal is to make the semi-supervised OT algorithm that can find a CM map that at the same time saves the class-wise structure using available labels in the target domain. Knowing the geometric property of OT maps, we proposed a novel DA problem statement for OT and a novel algorithm that improves OT's performance in semi-supervised settings.

$\textbf{1.2 The explanation of why this method works}$

OT is known as CM maps, and OT solvers implicitly use this property to find a CM map between domains. We build a new CM domain with respect to the target domain and save class correspondence between domains. Our method works because OT can find a CM map between target and source and at the same time save discriminativity, i.e., class-wise structure. OT maps the unlabeled samples to the perturbed samples from the same class in the source fiction domain because, in practice, the distance between samples inside the classes is less than the distance between samples from different classes.

$\textbf{1.3 Presentation issues}$

Thank you all for such a great and comprehensive review. We will improve the clarity of our paper and revise it according to comments.

References

[1] C. Villani. Optimal Transport: Old and New.   Grundlehren der mathematischen Wissenschaften. Springer Berlin Heidelberg, 2008.  ISBN 9783540710509.

---

### Decision · Program_Chairs · 2022-01-20

**Decision:**

Reject

**Comment:**

In this paper  propose a novel approach for semi-supervised domain adaptation based on the cyclic monotonicity property of optimal transport map. The main idea is to adapt (perturbed wrt a source classifier)  the labeled source samples  toward the target samples while preserving the known labels via the cyclical monotonicity. Then these perturbed samples can be used to perform classical OT domain adaptation. This pre-processing of the data has been shown in the numerical experiments to lead to better performance in average.

The proposed method has been found intriguing by all reviewers but the writing of the paper has been found clearly lacking and several suggestions were proposed by the reviewers. The choice of the authors to call the perturbed samples adversaries for instance made the paper harder to understand and the (anti-adversarial is also not a good choice of words). Another concern was that despite encouraging numerical results lack more baselines semi-supervised Domain Adaptation methods discussed by the reviewers were not compared (with or without OT).

The authors provided a short but clear response that was appreciated by the reviewers. But the clarifications promised by the authors were not done in the PDF during the editing period which means that the paper clearly needs a new round of reviews.  For this reason the consensus during the discussions was that this paper should be rejected. The AC believes that this is an interesting research direction that should be investigated but that the paper needs some more work before reaching the threshold for acceptance in selective ML venues. The authors are strongly encouraged to take into account the comments form the reviewers before resubmitting their work.